# Endocrine-Disrupting Chemicals and Their Effects during Female Puberty: A Review of Current Evidence

**DOI:** 10.3390/ijms21062078

**Published:** 2020-03-18

**Authors:** Laura Lucaccioni, Viola Trevisani, Lucia Marrozzini, Natascia Bertoncelli, Barbara Predieri, Licia Lugli, Alberto Berardi, Lorenzo Iughetti

**Affiliations:** 1Neonatal Intensive Care Unit, Department of Medical and Surgical Sciences of the Mothers, Children and Adults, University of Modena and Reggio Emilia, via del Pozzo 71, 41124 Modena, Italy; natascia.bertoncelli@gmail.com (N.B.); alberto.berardi@unimore.it (A.B.); 2Post Graduate School of Paediatrics, Department of Medical and Surgical Sciences of the Mothers, Children and Adults, University of Modena and Reggio Emilia, via del Pozzo 71, 41124 Modena, Italy; viola.trevisani@gmail.com (V.T.); luciamarrozzini@gmail.com (L.M.); lorenzo.iughetti@unimore.it (L.I.); 3Pediatric Unit, Department of Medical and Surgical Sciences of the Mothers, Children and Adults, University of Modena and Reggio Emilia, via del Pozzo 71, 41124 Modena, Italy; barbara.predieri@unimore.it

**Keywords:** puberty, Endocrine disrupting chemicals (EDCs), Window of susceptibility, Early Puberty, Estrogen-mimicking Endocrine Disruptors (EEDs), breast development, breast cancer

## Abstract

Puberty is the process of physical changes between childhood and adulthood during which adolescents reach sexual maturity and become capable of reproduction. It is considered one of the main temporal windows of susceptibility for the influence of the endocrine-disrupting chemicals (EDCs). EDCs may act as single chemical agents or as chemical mixtures; they can be pubertal influencers, accelerating and anticipating the processing of maturation of secondary sexual characteristics. Moreover, recent studies have started to point out how exposure to EDCs during puberty may predispose to breast cancer later in life. In fact, the estrogen-mimicking endocrine disruptors (EEDs) may influence breast tissue development during puberty in two main ways: the first is the action on the proliferation of the breast stromal cells, the second concerns epigenetic mechanisms. The aim of this mini-review was to better highlight what is new and what is not completely known regarding the role of EDCs during puberty.

## 1. Background

Endocrine-disrupting chemicals (EDCs) influence one or more functions of the endocrine system and cause several adverse health effects in human beings. They are ubiquitous in nature, being found in industrial chemicals, drugs, plastics, and other materials common in everyday life [1]. 

EDCs may represent a risk for our health, working through many different mechanisms of action. In fact, they may take several routes of absorption (oral, dermal, etc.), affect different temporal windows (prenatal and postnatal life, puberty), and be made up of many combinations of chemicals (mixtures), leading to extremely difficulty in interpreting their influence on human wellbeing. 

Moreover, their effect seems to be transgenerational, occurring over at least two or three generations. As the window of susceptibility, puberty is considered one of the hot spots in the lifetime when EDCs may exert their effects [2]. 

The main aims of this mini-review were (1) to provide an update of what we know about the relationships between EDCs and pubertal trends, and (2) to highlight the new role of EDCs as influencers of breast cancer predisposition later in life. 

## 2. Puberty and Pubertal Onset

Puberty is the process of physical change that occurs between childhood and adulthood, during which adolescents reach sexual maturity and become capable of reproduction. 

The main clinical manifestations of puberty are the growth spurt, progression of secondary sexual characteristics, development of gonads, and changes in body composition. These features lead to an increase in strength and endurance [3]. 

During pubertal development, changes in the brain activity are also detectable: the frontal lobes increase their volume and activity, leading to more complex psychological understanding [4]. The whole process involves years of development on both a mind and a body level.

In the last two centuries, pubertal timing has changed dramatically. Data show that the age of the menarche is experiencing a declining trend, from 17 years old at the beginning of nineteenth century to 13 years old by the 50s [5]. In the United States, the onset of puberty in girls is occurring earlier than previous studies have documented, with breast and pubic hair development appearing on average 1 year earlier in white girls and 2 years earlier in African-American girls. Beyond these data, it is also underlined how this early onset of puberty seems not to be correlated with a parallel anticipation of the menarche age. According to the most recent studies, the age of menarche has decreased by only few months [6].

As far as timing is concerned, in the 60s, Tanner et al. identified the normal range of pubertal onset as between 8 and 13 years in girls, and between 9 years and 6 months and 13 years and 8 months in boys [3,4]. Therefore, the development of secondary sexual characteristics is considered precocious when it happens in girls under the age of 8 years old and in boys younger than 9 years old. 

The timing of pubertal onset depends on genetic and environmental factors. The genes involved in this process are the ones that control GnRH hypothalamic secretion, pituitary development and its functioning, hormone synthesis and bioactivity, energy homeostasis and growth, and potential peripheral feedback from sex steroids [7]. 

Studies show that some populations are much more likely to develop precocious puberty: girls; African-Americans; individuals presenting other medical conditions (McCune–Albright syndrome, congenital adrenal hyperplasia) or experiencing or having experienced radiation therapy of the central nervous system; those exposed to sex hormones; children that may have been in contact with estrogen or testosterone cream or ointment, or with other substances such as adults’ medications, dietary supplements, or food containing phytoestrogens.

The nutritional status of the pediatric population influences the age of pubertal onset. Adolescents with anorexia nervosa usually reach menarche later [8], highlighting a correlation between nutritional state and development. Girls who are overweight or obese have a higher probability of being diagnosed with central precocious puberty or early puberty compared to the general population [9]. Genomic studies and human epidemiological data suggest a correlation between early infancy weight gain, early pubertal onset, and later obesity in adulthood, which can be explained by the overlapping of genes involved in pubertal onset and obesity [10]. Moreover, obesity induces hyperinsulinemia and hyperleptinemia, which can affect linear growth and advanced puberty [11]. 

Both events, pubertal timing and obesity, are influenced by the environment, and there is a strong interaction between genes and environments during critical periods of development such as puberty. This interaction is due to substances, EDCs, which derail the normal hormonal process. 

## 3. EDCs and Puberty

In 2015, the Endocrine Society defined an EDC as “an exogenous chemical or mixture of chemicals, which interferes with any aspect of hormone action”. The EDCs affect the control of energy balance, reproduction, hormone-sensitive cancer, thyroid, neurodevelopment, and the neuroendocrine system. They can act at any level in the hypothalamic–pituitary–gonadal system or the peripheral tissue. During a lifespan, there are some critical points (windows of development) that are important because during them, cells are rapidly proliferating and epigenetic changes are more likely to occur. These periods are fetal life, neonatal life, and puberty [12].

Moreover, EDCs act at any level in the hypothalamic–pituitary–gonadal–peripheral tissue–endocrine axis. Indeed, they can act not only as agonists or antagonists of estrogen receptors, but can also target and act against androgen and progesterone receptors. They may also mimic the natural occurrence of estrogens and androgens. Furthermore, EDCs bind to cell receptors and block the functions of endogenous hormones, thus acting as anti-estrogens and anti-androgens [13].

The most important effect of EDCs is considered to be the anticipation of or, more generally, impact on the timing of female puberty [14]. Nevertheless, this effect is difficult to demonstrate since low-level exposures to a mixture of EDCs may mask the effects of some of them. Moreover, inter-individual variation in susceptibility might interfere with an objective analysis. Finally, the effects on humans to EDCs exposure may show up with a considerable delay.

We analyzed the most important and well-known EDCs that might interfere with natural pubertal timing (Table 1).

### 3.1. BPA

Bisphenol A (BPA) is a precursor of plastics, polycarbonates, and epoxy resins. BPA-based plastic is clear and tough, and it is used in a lot of common objects such as plastic bottles, food storage containers, baby bottles, and CDs. Epoxy resins are utilized to line water pipes, as coating on the insides of many food and beverage containers, and in making thermal paper. This chemical is almost ubiquitous, which means that people are at great risk for exposure and, even if the estrogen receptor agonist activity is weak, its potential should not be underestimated. BPA also acts as an anti-androgen.

Studies demonstrate how this EDC is associated with premature thelarche, precocious puberty and pubertal development [15,16,17].

### 3.2. DDT/DDE

Dichlorodiphenyltrichloroethane, commonly known as DDT, is an organochlorine. It was originally developed as an insecticide for use in agriculture and it is colorless, tasteless, and odorless. Because of these features, exposure to DDT is imperceptible, but being exposed during fetal life and lactation can affect sexual development. Even if DDT is banned from the market, it will remain present in the environment as a persistent organic pollutant (POP). Moreover, in some low-income countries, it is still widely used.

DDT and its metabolite dichlorodiphenyldichloroethane (DDE) are the most studied of the pesticides that have been identified as EDCs. Both have estrogenic, anti-androgenic, and antiprogestin effect and induce aromatase. According to Vasiliu et al. (2004), there is an association between exposure to these chemicals and precocious puberty, delayed puberty, and earlier age of menarche [18,19].

### 3.3. Dioxins

Dioxins are byproducts of burning or various industrial processes such as chlorinated herbicide production, smelting, or chlorine bleaching of paper. They have an estrogenic and anti-androgenic effect, and they interfere with sex steroid synthesis. Exposure to dioxins during the prenatal and lactation period is associated with delayed breast development. On the other hand, exposure to these chemicals has not been correlated with age of menarche or pubic hair development [20]. 

### 3.4. PBDEs and PBB

Flame retardants are a group of chemicals added to manufactured materials (plastics, textiles, surface finishes, and coatings). They are intended to prevent or slow down the further development of ignition. Some examples of such chemicals are polybrominated diphenyl ethers (PBDEs) and polybrominated biphenyl (PBB). They act as estrogens and anti-androgens. Data show an association between PBDEs and premature thelarche, earlier menarche, and earlier pubertal development [21,22]. Furthermore, prenatal exposure to PBDEs is associated with later menarche in girls and earlier development of pubic hair in boys [23].

Moreover, exposure during the peri-pubertal period seems to interfere with reproductive development [24]. 

### 3.5. PCBs

Polychlorinated biphenyl (PCBs) is a dioxin-like compound derived from biphenyl. This chemical is widely used as a dielectric and coolant fluid in electrical apparatuses, in carbonless copy papers, and in heat transfer fluids. Its mechanism of action is rather similar to that of dioxins, but compared to dioxins, there is evidence that exposure during the prenatal period leads to early onset of menarche and to delayed pubertal development [25].

### 3.6. Phthalates

Phthalates are esters of phthalic anhydride. They are used as liquid plasticizers in plastic, flooring, personal care products, medical devices, and tubing. They increase the flexibility, transparency, durability, and longevity of materials. Their most common use is to soften polyvinyl chloride (PVC).

Their endocrine-disrupting mechanism is not fully clarified, but they either act as estrogen receptor agonists and antagonists, or androgen receptor antagonists. Moreover, they can also disrupt androgen synthesis.

Different studies have demonstrated a significant association with premature thelarche and precocious or early puberty [26,27].

Phthalates can be classified as low- or high-molecular-weight phthalates. Depending on the class and the timing of exposure, different outcome have been observed. High-molecular-weight phthalate levels several years before puberty are associated with later pubic hair development and younger age of menarche; in contrast, low-molecular-weight phthalate levels are related to advanced breast or pubic hair development [28,29].

### 3.7. Pyrethroids

A pyrethroid is an organic compound similar to a natural pyrethrin, which comes from the pyrethrum flower. This chemical is found in most common commercial household insecticides. It has been reported both as a weak androgen and for its association with late onset of puberty in girls [30,31].

## 4. Puberty and EDCs: A Temporal Window for Breast Cancer

In girls, the first physical sign of puberty is thelarche, with physical breast changes in both the stroma and the epithelium. The amount of fibrous and fatty tissue in the stroma increases (the adult non-lactating breast is more than 80% composed of stroma) [32]. The ducts first elongate, proliferate, and form lobular structures composed of the growth ductal tree; with the achievement of sexual maturity, they decrease in size, the proliferative phase reducing [33]. Thus, puberty is the time in which the breast tissue undergoes rapid changes regulated by small fluctuations of endogenous hormones [34]. These rapid structural and cellular modifications are thought to create a “window of susceptibility” [35].

Breast tissue during this period of morphogenesis and re-modeling is particularly sensitive to carcinogenic effects; studies have demonstrated that exposure to radiation, DDT, medications, and disease before and during puberty increases breast cancer risk in adulthood [36,37,38,39,40].

The role of estrogen in breast development and tumorigenesis is confirmed. Prolonged and uninterrupted exposure to endogenous estrogen and administration of exogenous estrogens to postmenopausal women increases the risk of cancer. In contrast, gonadectomy in premenopausal women reduces the risk of breast cancer, while pharmacological agents serve to treat the same condition, blocking estrogen biosynthesis or estrogen receptors.

It is not entirely clear how human exposure to specific EDCs can be linked to breast cancer. The EDCs of interest in this mechanism are those with estrogenic activity: they are called estrogen-mimicking endocrine disruptors (EEDs).

The main effects of EEDs are two-fold: the first is their action on the proliferation of the stromal cells, the second concerns epigenetic mechanisms. EEDs affect stromal cells, interfering with the estrogen signal pathway [40,41].

Moreover, EEDs may modify the breast matrix composition. All these changes macroscopically create a different breast density, which is one of the strongest predictors of premenopausal breast cancer risk [42].

Breast cancer is also associated with epigenetic changes such as DNA methylation, histone modifications, and non-coding RNAs (ncRNAs). These mechanisms work together to determine whether the epigenome may expresses itself and how, facilitating gene expression control, X chromosome inactivation, and/or genomic imprinting [43].

The most important chemical EEDs and their activities to promote or facilitate breast cancer are analyzed below (Table 1).

### 4.1. Atrazine

Atrazine is an herbicide used to prevent pre- and post-emergence broadleaf weeds in crops (corn and sugarcane) and turf. It can induce aromatase activity, increasing the level of estrogen in the body. Atrazine is not classified as a carcinogen or mutagen because its link with breast cancer is not totally clear. On this issue, studies are contradictory. Some studies on human reaction to atrazine exposure have seemed to detect a weak association between atrazine and breast cancer [44].

### 4.2. BPA

BPA increases the methylation levels of key genes associated with tumor development (e.g., BRCA1, CDKN2A, etc.), altering the epigenome in order to promote proliferation, senescence, and tumor development [45].

3D cultures of breast cells have shown that BPA reduces the number of tubules, increases spherical masses, and causes more deformed acini, indicating an ability to induce neoplastic transformation [46]. This because BPA exposure increases HOXB9 expression in breast cells, both in vitro and in vivo, through a mechanism that involves increased recruitment of transcription and chromatin modification factors. HOXB9 is a homeobox-containing gene that plays a key role in mammary gland development, and is associated with breast cancer. BPA acts as a competitor for estrogen receptors (ERs), which leads to the activation of this gene through chromatin modifications (histone methylation and acetylation) [47]. 

### 4.3. DDT/DDE

The estrogenic property of these compounds and their prominent role in mRNA regulation increase epithelial cells’ proliferation index and mammary tumor cell proliferation [48]. Moreover, a prospective study showed that girls exposed to high levels of DDT and DDE were more likely to develop breast cancer than those who were exposed to a lower dose. On top of that, the younger the age at which a person was exposed to these compounds, the higher the risk of developing breast cancer [38]. These data support the hypothesis that the age of exposure must be considered for risk assessment.

### 4.4. DES

Diethylstilbestrol (DES) is a nonsteroidal estrogen medication that was used in the past to support during pregnancy women with a history of recurrent miscarriage, as hormone therapy for menopausal symptoms and estrogen deficiency, and as a treatment for breast cancer.

The development of breast cancer in women exposed to DES is mediated by the proliferative effects of estrogens, and by a differential expression of ncRNAs. In particular, epigenetically silenced mRNA causes disordered regulation of the p53-mediated apoptotic pathway [49].

### 4.5. Parabens

Parabens are a widely used class of preservatives in cosmetic and pharmaceutical products. They are used for their bactericidal and fungicidal properties. They are commonly found in shampoos, commercial moisturizers, topical/parenteral pharmaceuticals, suntan products, make-up, and toothpaste, and are also used as food preservatives.

They have estrogen-like effects and they may be associated with breast cancer etiology. This is demonstrated by the association between the utilization of paraben-containing body care products in the Western world and the increased incidence of breast cancer [50].

### 4.6. PCBs

Studies have demonstrated that polychlorinated biphenyls accumulate in breast adipose tissue, but so far, an association has not been proven between PCB exposure and breast cancer [51]. This is probably because PCBs are mixtures of congeners with varied properties, and because breast cancers of different subtypes are etiologically distinct diseases. The absence of stratified subgroup analysis on individual PCB exposure and patients with specific biological subtypes of breast cancer may have resulted in an underestimation of the correlations between PCBs and breast cancer [52]. Moreover, PCBs’ mechanisms of action are different. In fact, the natures of the biochemical and toxic effects of PCB congeners are largely determined by their structure, as well as different PCB congeners sometimes exerting conflicting actions. Some PCBs act as estrogen agonists both in vitro and in experimental animal systems; some others have been shown to be anti-estrogenic, while others are enzyme inducers of phenobarbital (PB)-type cytochrome P450 [53]. The main mechanism involved in breast cancer seems to be the estrogen agonist mechanism.

### 4.7. PFOA

Perfluorooctatonic acid (PFOA) serves as a surfactant in the emulsion polymerization of fluoropolymers and as a building block for polymeric materials. It is used for various industrial applications (carpeting, upholstery, floor wax, firefighting foam, sealants etc.). Data show how in CD-1 and C57B1/6 mice, early PFOA exposure alters the mammary gland without changing other pubertal endpoints [54]. This development may increase susceptibility to carcinogenesis in future generations.

### 4.8. Phthalates

One characteristic of phthalates is their lipophilicity, so they accumulate in breast tissue where they cause modification of the mammary gland [55]. Epigenetic changes due to exposure to phthalates increase the proliferation of cells in the breast [56].

### 4.9. TCDD

2,3,7,8-Tetracloridibenzo-p-dioxin, or TCDD, is a famous organochlorine mainly known as a contaminant in Agent Orange, an herbicide used in the Vietnam War. The scientific community is aware of its catastrophic effects, including birth defects, pregnancy loss, and increased rates of cancer. In 2012, a retrospective study showed a significantly higher prevalence of breast cancers in women working in a chemical pesticide establishment due to their exposure to high levels of TCDD [57].

TCDD has a significant effect on the epigenome in breast and other tissues. It is interesting to note that TCDD induces an epigenetic silencing of key tumor suppressors, which may have a role in breast cancer development [58].

### 4.10. Vinclozolin

Vincozolin is a common dicarboximide fungicide used to control diseases in vineyards and on fruits and vegetables. Exposure to this pesticide in female rats causes the formation of mammary tumors and dysregulates gland development [59]. It promotes an epigenetic alteration in the germ line that creates a transgenerational disease state [60].

## 5. Conclusions and Perspectives

Most of these EDCs are present in everyday objects and in various common solutions; it is almost impossible not to be exposed to these chemicals, even if laws were to ban the use of most of them and restrict the others. It is known that environmental factors, including EDCs, are the most important determinant in explaining the puberty secular trend and the timing of menarche. The anticipation of these two endpoints of development can be explained by the action of environment during the pubertal period, and how it may advance and activate the central axis.

According to Parent et al., this paradigm of EDCs could be revised; in fact, environmental factors can affect puberty and reproduction through central and peripheral mechanisms, taking advantage of the critical windows of development [61]. During these periods, EDCs can modify genetic or epigenetic pathways.

The epigenetic pathways are interesting because of their long-term action. Studies on animals have demonstrated the effects of a mixture of EDCs on female sexual development over three generations, and that the effects are even bigger in the second and third generations (delayed vaginal opening, decreased percentage of regular estrous cycle, and decreased GnRH interpulse interval) than in the first. The reproductive phenotypes in rats are associated with alterations in both transcriptional and histone posttranslational modifications of hypothalamic genes involved in reproductive competence. Moreover, females of the first generation exposed in utero to these EDCs showed alteration of the thyroid and impaired maternal behavior, which could cause additional epigenetic change in subsequent generations [62].

EDCs can also act as obesogens and promote early adiposity rebound, changing metabolic or peripheral signals, and increases in adrenal androgen levels, inducing early pubertal development [63].

EDCs, and in particular EEDs, may produce epigenetic changes not only in puberty genes but also in other tissues and cells. In fact, these compounds could act as carcinogens and induce breast cancer. Clearly, EEDs are not the only factors responsible for these phenomena. Their effect is influenced by environmental carcinogens and unhealthy foods such as refined sugar, processed fats, and food additives, which promote molecular damage that increases the risk of breast cancer.

Puberty amplifies this process; young women today live in an obesogenic world and they are exposed to several contemporaneous risk factors, especially girls who live in disadvantaged conditions [64].

Moreover, the exposure to these compounds and the development of mammary glands expands the epigenetic changes that EEDs cause in breast tissue, promoting tumorigenesis and increasing the chance of damage that might prove to be significant.

In conclusion, exposure to EDCs during puberty increases the risk of developing breast cancer during the lifetime.

Although these results are still somewhat controversial, it is important for pediatricians to keep in mind the new risks for young girls during puberty, and to provide the right information to their families.

## Figures and Tables

**Table 1 ijms-21-02078-t001:** Description of several EDCs and their effects of puberty and breast cancer predisposition.

EDCs	Sources	Half-Life	Exposure	Critical Window Exposure	Mechanism	Effect on Puberty	Effects on Breast
**Atrazine**	Herbicide	<1 day	Dermal absorption, inhalation	Prenatal life	ER agonist	Early puberty	↑incidence of breast cancer
**BPA**	Polycarbonate plastics, epoxy resins, plastic toys and bottles, lining of food cans	4–5 h	Ingestion, dermal absorption, inhalation	Prenatal life Puberty	ER agonist Antiandrogen	Premature thelarche	Induce neoplastic transformation
**DES**	Nonsteroidal estrogen medication used in the past	3–6 h	Ingestion	Prenatal life Puberty	ER agonist Epigenetic silencing of mRNA	Not known	↑breast cancer
**DDT/DDE**	Contaminated water, soil crops, fish	6–10 years	Ingestion, dermal absorption, inhalation	Prenatal life and lactation period	ER agonist Antiandrogen Antiprogestin Induction of aromatase	Precocious puberty Anticipated menarche Later onset of puberty	↑cells proliferation index and mammary tumor cells proliferation
**Dioxin**	By-product of chlorinated herbicide production, smelting, chlorine bleaching of paper	7–11 years	Ingestion, inhalation	Prenatal and lactation Puberty	ER agonist Antiandrogen Interfere with sex-steroid synthesis	Delayed breast development	Induce epigenetic silencing of key tumor suppressor (TCDD)
**Parabens**	Cosmetics and pharmaceutical products, toothpaste and food preservatives.	<24 h	Dermal absorption, ingestion	Pre- post-natal life and puberty	ER agonist	Not known	↑incidence of breast cancer
**PBDEs/PBB**	Furniture, mattresses, carpet pads, automobile seats, flame-retardant textiles	2 days–3 months	Ingestion, dermal abruption, inhalation	Prenatal life, perinatal period, peripuberty	ER agonist/antagonist Antiandrogen	Early pubic hair in boys Premature thelarche Early/late menarche Early puberty	Not known
**PCBs**	Contaminated air and food, skin contact with old electrical equipment	12 days–16 years	Ingestion, dermal absorption, inhalation	Prenatal life	ER agonist/antagonist Antiandrogen	Early onset of menarche Delayed pubertal development	Accumulates in breast adipose tissue
**PFOA**	Carpeting, upholstery, floor wax, firefighting foam, sealants	3.8 years	Ingestion, inhalation	Prenatal life Puberty	Induced estrogen	Delayed pubertal development	↑susceptibility to carcinogenesis in future generations
**Phthalates**	Contaminated food, PVC plastics and flooring, personal care, medical devices and tubing	12 h	Ingestion, dermal absorption, inhalation	Prenatal life	ER agonist/antagonist Antiandrogen Interfere with androgen synthesis	Early puberty Premature thelarche Delayed pubic hair development	↑cells’proliferation in the breast
**Pyrethroids**	Contaminated water, soil, food	10 h	Ingestion, dermal abruption, inhalation		Androgen agonist	Delayed puberty	Not known
**Vinclozolin**	Fungicide	1–3 months	Ingestion, dermal absorption, inhalation	Prenatal life Puberty	ER agonist Antiandrogen Epigenetic alteration	Not known	Dysregulates the gland development ↑formation of mammary tumor

↑: Increased incidence.

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
