# Peer review of "Endocrine-Disrupting Chemicals and Their Effects during Female Puberty: A Review of Current Evidence"

_ijms, 2020, doi:10.3390/ijms21062078_

Round 1

Reviewer 1 Report

This is a useful review of endocrine disruptors in puberty focused mainly on pubertal women. The title of the paper is poor English and doesn't do justice to the paper. The authors might consider changing the title such as Endocrine Disrupting Chemicals and effects during puberty: a review of current evidence.

My only criticism is there is little discussion about pubertal boys and for example the effect on spermatogenesis. It is believed, for example, that EDCs are at least partly responsible for declining sperm counts in USA and Europe.

Minor points

  • the EDCs are discussed in random order- I might have expected them to be arranged in alphabetical order starting with BPA and finishing with Vinclozolin.

Several places where the English could be improved..

  • l20 p1 exposure rather than exposition
  • l38 p1 hot spots rather than hot spot
  • l41 p1 influencers rather than influencer
  • l65 p2 that rather than who
  • l69 p2 conditions rather than condition
  • l93 p2 please define HPG
  • l131 p8 mRNA rather than miRNA

Author Response

We would like to thank the reviewer for his/her useful comments.

We have changed the manuscript according to your requests.

First of all, we changed the title as suggested.

We have also specified in the title that the work regards puberty in female. We do agree with the reviewer that there are not enough references and sentences for male, looking at important aspects, as fertility or development of external genitalia. 

Moreover, we changed language as suggested and we specified the EDCs in alphabetic order both in the text and in the table. 

References list has changed, accordingly. 

Reviewer 2 Report

This is an important elaboration on effects of endocrine disrupting chemicals on women breast and reproductive system physiology. Some improvements are suggested:

1. Minor English correction e.g.
Avoid noun variety „menarche’ ” „scells’ ”
Correct „DDT e DDE”

2. More delatils on mechanism, factors, receptors, genes, signaling pathways etc. involved in environmenrtal chemical action is needed e.g.
-give more details „3D cultures of breast cells show how BPA reduces the number of tubules, increases spherical masses, and causes more deformed acini, indicating an ability to induce neoplastic transformation 100 [45].”
„As we already 120 mentioned, what is known is that PCBs interfere with estrogen metabolism, so they are likely to influence sex steroid biosynthesis in the breast via epigenetic mechanism.”

Author Response

Dear Reviewer, thanks for you kind judgment on our paper. 

Attached is the revised version of the manuscript according to your suggestions. 

1. We have made linguistic changes as you said, in particular we avoided the noun variety and corrected "DDT e DDE". 

2. We gave more details for BPA and PCBs and breast cancer as requested.